# Efficacy of AI-Assisted Personalized Microbiome Modulation by Diet in Functional Constipation: A Randomized Controlled Trial

**DOI:** 10.3390/jcm11226612

**Published:** 2022-11-08

**Authors:** Naciye Çiğdem Arslan, Aycan Gündoğdu, Varol Tunali, Oğuzhan Hakan Topgül, Damla Beyazgül, Özkan Ufuk Nalbantoğlu

**Affiliations:** 1Department of General Surgery, School of Medicine, Medipol University, Istanbul 34214, Turkey; 2Department of Microbiology and Clinical Microbiology, Faculty of Medicine, Erciyes University, Kayseri 38280, Turkey; 3Department of Emergency Medicine, Eşrefpaşa Municipality Hospital, Izmir 35170, Turkey; 4Department of Parasitology, Faculty of Medicine, Celal Bayar University, Manisa 45040, Turkey; 5Enbiosis Biotechnology, Istanbul 34398, Turkey; 6Department of Computer Engineering, Faculty of Engineering, Erciyes University, Kayseri 38280, Turkey

**Keywords:** functional bowel disorders, gut microbiota, personalized diet, machine learning, personalized medicine, Turkey

## Abstract

Background: Currently, medications and behavioral modifications have limited success in the treatment of functional constipation (FC). An individualized diet based on microbiome analysis may improve symptoms in FC. In the present study, we aimed to investigate the impacts of microbiome modulation on chronic constipation. Methods: Between December 2020–December 2021, 50 patients fulfilling the Rome IV criteria for functional constipation were randomized into two groups. The control group received sodium picosulfate plus conventional treatments (i.e., laxatives, enemas, increased fiber, and fluid intake). The study group underwent microbiome analysis and received an individualized diet with the assistance of a soft computing system (Enbiosis Biotechnology^®^, Sariyer, Istanbul). Differences in patient assessment constipation–quality of life (PAC-QoL) scores and complete bowel movements per week (CBMpW) were compared between groups after 6-weeks of intervention. Results: The mean age of the overall cohort (*n* = 45) was 31.5 ± 10.2 years, with 88.9% female predominance. The customized diet developed for subjects in the study arm resulted in a 2.5-fold increase in CBMpW after 6-weeks (1.7 vs. 4.3). The proportion of the study group patients with CBMpW > 3 was 83% at the end of the study, and the satisfaction score was increased 4-fold from the baseline (3.1 to 10.7 points). More than 50% improvement in PAC-QoL scores was observed in 88% of the study cohort compared to 40% in the control group (*p* = 0.001). Conclusion: The AI-assisted customized diet based on individual microbiome analysis performed significantly better compared to conventional therapy based on patient-reported outcomes in the treatment of functional constipation.

## 1. Introduction

Constipation is a common gastrointestinal disorder with an estimated global prevalence of 14% [1] and represents a heavy burden for ambulatory healthcare systems [2]. Chronic constipation is defined as difficult and/or infrequent bowel movements and is divided into four subgroups: functional constipation (FC), irritable bowel syndrome (IBS) with constipation, opioid-induced constipation, and functional defecation disorders [3]. Among these, FC has been the least understood and the most desperate group, as only one-third to half of the patients benefit from available treatments [4,5]. Similarly to some common comorbidities, quality of life (QoL) is impaired [6]. The impact of FC is estimated to cause a mean loss of 2.4 active days in a month [7]. Moreover, both direct and indirect healthcare costs are determined by approximately 2.5 million visits and 92,000 hospitalizations per year, with more than 7 billion USD for diagnostic assessments [8,9].

The current guidelines on the diagnosis and management of constipation in adults recommend the symptomatic approach as the initial step [10]. First-line treatments include changes in lifestyle and diet, cessation of medications causing constipation, fiber and/or bulk-forming agents, increased fluid intake, and exercise. The second step includes laxatives, and the third step is the introduction of stimulant laxatives, enemas, as well as prokinetic drugs [10]. In a recent meta-analysis, the results of 33 studies involving 17,214 patients, revealed that almost all medications were superior to placebo in terms of achieving three or more complete bowel movements per week (CBMpW) and the diphenylmethane laxatives (prucalopride and sodium picosulfate) ranked as the most effective [4]. As most of the studies in the literature report results after 4–12 weeks, the long-term effects of the medications and the sustainability of the treatments have been a main topic of debate [4,5,11]. Besides, the main reasons for dissatisfaction with medications are low efficacy and the fact that half of the patients have reported concerns about adverse effects with long-term use [12]. The ‘symptomatic approach’ rationale of available options and the lack of any radical treatments justify these concerns.

In recent studies, it has been observed that the intestinal microbiota in patients with FC is different from that of healthy individuals [13]. Although the role of the microbiome in CC pathophysiology is not yet fully understood, it is suggested that gut microbiota may have modulating effects on gastrointestinal motility or metabolites, and fermentation products may cause increased gas formation [13]. Animal studies revealed that colonization of germ-free mice with microbiota increased the encoding of several proteins (L-glutamate transporter, L-glutamate decarboxylase, g-aminobutyric acid, vesicle-associated protein 33, enteric g-actin, and cysteine-rich protein 2) which have neuromodulator effects on the enteric nervous system [12]. Human studies have also indicated the crucial role of the microbiome in gastrointestinal motility. An increased proportion of *Actinobacteria*, *Bacteroides*, *Lactococcus*, and *Roseburia* are associated with faster gut transit time, whereas Faecalibacterium correlates with slower motility [12]. The present study aimed to investigate the impact of an AI-assisted microbiome-based personalized diet compared with sodium picosulfate plus conventional therapy (i.e., laxatives, enemas, increased fiber, and fluid intake) on FC patients.

## 2. Materials and Methods

The study was approved by the Institutional Ethics Committee (Approval no. 10840098-772.02-E.47859) and conducted in line with the Declaration of Helsinki. The patients were thoroughly informed about the protocol, and written consent was obtained. Patients fulfilling the Rome IV criteria for FC and aged between 20–65 years were included in the study. All the patients underwent detailed physical and rectal examinations by a European board-certified coloproctologist (NCA). Patients who had a colonoscopy performed within the last 5 years were included. Colonic transit time and magnetic resonance defecography were obtained from all patients. Exclusion criteria were: the use of antibiotics, probiotics, and/or prebiotics within the last four weeks; gastrointestinal endoscopy within the last four weeks; a history of major gastrointestinal surgery (total/segmental gastrectomy, small bowel resection, and/or colonic resection); cholecystectomy; inflammatory bowel diseases; and celiac disease. Any etiology of chronic constipation other than FC (irritable bowel syndrome, rectocele, dyssynergic defecation, and opioid use) was excluded. Patients with endocrine, metabolic, or neurologic disorders causing constipation (hypothyroidism, Parkinson’s disease, and paraplegia) were also excluded from the study.

### 2.1. Study Design and Groups

This was a single-center, prospective, randomized study. Patients were those who consulted with the Istanbul Medipol University Hospital General Surgery Clinic with constipation. Patients fulfilling inclusion criteria were divided into two groups using block randomization at a 1:1 ratio. The coloproctologist (NCA) was not blinded to randomization as she obtained the fecal samples from the patients in the study group and managed the treatments of the control group. Baseline and post-treatment questionnaires were collected by another surgeon blinded to the randomization (OHT).

After randomization, both groups were recommended to continue their regular diets with increased fluid and fiber intake and informed about the exclusion criteria. The control group received 5 mg of sodium picosulfate (Dulcolax^®^ 2.5 mg, Sanofi, Turkey) daily for ten weeks. In the study group, after fecal samples were taken, patients were suggested to continue their regular diet for four weeks until the microbiome analysis was completed. During the subsequent six weeks, patients in the study group received the personalized microbiome modulatory diet, and those in the control group received 5 mg of sodium picosulfate plus the conventional treatments (i.e., laxatives, enemas, increased fiber, and fluid intake) for FC. The two groups were compared in terms of bowel movements and quality of life.

The primary endpoint was the proportion of patients with a mean of three or more complete bowel movements per week (CBMpW) at ten weeks. The secondary endpoint was a more than 50% improvement in the total Patient Assessment Constipation Quality of Life (PAC-QoL) score.

### 2.2. Fecal Sampling and 16S Ribosomal RNA Gene Sequencing

Fecal samples were collected using BBL culture swabs (Becton, Dickinson and Company, Sparks, MD, USA) and transported to the laboratory in a DNA/RNA shield buffer medium. DNA extraction was carried out directly from the stool samples using a Qiagen Power Soil DNA Extraction Kit (Qiagen, Hilden, Germany). A NanoDrop (Shimadzu, Japan) device was used to measure the final concentrations of extracted DNA. dsDNA quantification was done using the Qubit dsDNA HS Assay Kit and a Qubit 2.0 Fluorimeter (Thermo Fisher Scientific, Waltham, MA, USA).

The sequencing of 16S rRNA was performed using the Illumina MiSeq (Illumina, San Diego, CA, USA) device according to the manufacturer’s protocol.

All amplified products were then checked with 2% agarose gel electrophoresis. Amplicons were purified using the AMPure XP PCR Purification Kit (Beckman Coulter Genomics, Danvers, MA, USA) and quantified using the Qubit dsDNA HS Assay Kit and a Qubit 2.0 Fluorimeter (Thermo Fisher Scientific, Waltham, MA, USA). Approximately 15% of the PhiX Control library (v3) (Illumina, San Diego, CA, USA) was combined with the final sequencing library. The libraries were processed for cluster generation. Sequencing on 250PE MiSeq runs was performed, generating at least 50,000 reads per sample.

Sequencing data were analyzed using the QIIME pipeline [14] after filtering and trimming the reads for a PHRED quality score of 30 via the Trimmomatic tool [15]. Operational taxonomic units were determined using the Uclust method, and the units were assigned to taxonomic clades via PyNAST using the Green Genes database [16] with an open reference procedure. Alpha- and beta-diversity statistics were assessed accordingly by QIIME pipeline scripts. The graph-based visualization of the microbiota profiles was performed using the tmap topological data analysis framework with the Bray-Curtis distance metric.

### 2.3. The AI-Based Personalized Nutrition Model

The AI-based nutritional recommendations system is based mainly on the eating rates of the individual in a certain period to ensure the homeostasis of the microbiome and increase microbial diversity.

After the analysis reports are released, a detailed health-disease life history is taken, and a six week diet service is provided to the individual with lifestyle-specific diet lists in accordance with his/her comorbidities. Diet lists are updated according to the individual’s feedback, recovery level, and wishes during weekly meetings.

While designing an individual’s diet list, the modules in the Microbiome Analysis Report provide detailed data and help design results-oriented diet lists. In this study, foods containing “fiber” were prioritized in the AI-based recommended food scores specific to constipated individuals and integrated into the diet list in accordance with the individual’s lifestyle. The Enbiosis personalized nutrition model estimates the optimal micronutrient compositions for a required microbiome modulation. The present study computed the microbiome modulation needed for a constipated patient based on the “constipation” indices generated by the machine learning models as described previously [17]. While designing the diet lists, care was taken not to give calories below the basal metabolic rate.

### 2.4. Assessments and Follow-Up

Demographic and clinical characteristics, as well as the number of CBMpW and PAC-QoL scores of eligible patients, were recorded at baseline. The PAC-QOL questionnaire was previously validated in the Turkish population and assesses constipation-related symptoms on four subscales (physical discomfort, psychosocial discomfort, worries and concerns, and satisfaction) that are scored on a 5-point Likert-type scale (0, none/not at all; 4, extremely/all the time) and are inversely proportional with symptom relief [18]. All the patients were asked to record daily defecation diaries, which include the frequency of bowel movements, presence of straining and/or feeling of incomplete evacuation, and/or use of any rescue enema. The diaries were collected, and PAC- QOL questionnaire was repeated at 10 weeks. The absence of more than 2 weeks of diary records was defined as ‘non-responders’. For less than 2 weeks of absent data, the information from last week was copied for the missing weeks.

According to the microbiome test results, patients in the study group received AI-assisted, personally customized diets (Enbiosis Biotechnology^®^, Sariyer, Istanbul, Turkey) with weekly online dietitian support for six weeks.

### 2.5. Statistical Analysis

A successful treatment and patient satisfaction rate of 30% was estimated with conventional treatments of FC [19]. With the hypothesis that soft-computed microbiome treatment would increase CBMpW to ≥3 in 80% of the patients, the sample size was calculated as 19 patients in each group with α = 0.05 and 90% power. Considering a drop-out rate of 25%, a total of 50 patients were recruited for the study. Power and sample size analyses were performed by a web-based software (Raosoft Inc., Seattle, WA, USA) [20].

Continuous variables were expressed as means and standard deviation, and categorical variables as frequency and percentages. The distribution of continuous variables was determined by histograms, skewness, and Kurtosis analyses. The association between parametric variables was tested by an independent samples *t*-test. The association between non-parametric variables was determined by Mann-Whitney-U. Differences in mean CBMpW and PAC-QoL scores before and after treatments were tested by a paired-samples *t*-test. The difference between categorical variables was tested by a *chi*-square test. Statistical significance was defined as *p* < 0.05. Statistical analyses were performed using SPSS 21.0 (IBM, Chicago, IL, USA).

## 3. Results

Between December 2020 and December 2021, 74 patients with constipation were assessed for eligibility, and 50 were randomized into control (*n* = 25) and study (*n* = 25) groups, yet 5 patients in the control group were excluded for various reasons. The flow diagram is given in Figure 1. The mean age was 31.5 ± 10.2, and 40 (88.9%) patients were female. The mean age in the control group was 34.5 ± 11.4, which higher than the study group mean age (29.1 ± 8.6), but this difference was not statistically significant (*p* = 0.076). Four (8.9%) of the patients had comorbidities including type 2 diabetes (*n* = 2), asthma (*n* = 1), and hypertension (*n* = 1); 10 (22.2%) had proctologic diseases (3 anal fissures and hemorrhoids). The mean duration of constipation was 88.8 ± 66.9 months. The baseline CBMpW was ≥3 in 6 (13.3%) of the patients, with a mean value of 1.9 ± 1.92. There was no difference between the groups in terms of gender, body mass index, duration of constipation, or stool frequency (Table 1). The mean baseline PAC-QoL score was 55.3 ± 14.6 and was similar between the groups (*p* = 0.101), except for psychosocial discomfort. The mean scores of PAC-QoL subscales were not different between groups at baseline (Table 1).

After 10 weeks, the mean CBMpW improved from 2.1 ± 2.2 to 2.8 ± 2 in the control group (*p* = 0.003) and from 1.7 ± 1.6 to 4.3 ± 1.8 in the study group (*p* > 0.001). The mean total PAC-QoL scores improved in both groups. There was a slight but significant improvement in the control group (59.3 ± 10.4 to 55 ± 8.5, *p* = 0.005)) and an approximately 3.5-fold significant improvement in the study group (52.1 ± 16.9 to 15.9 ± 16, *p* = 0.001). Among PAC-QoL subscales, only worries and discomfort scores improved after treatment in the control group, whereas the study group has significantly improved scores in every measure (Table 2).

The mean post-treatment CBMpW was lower than 3 and significantly lower in the control group compared to the study group (2.8 ± 2 vs. 4.3 ± 1.8, *p* = 0.013). In every measure of PAC-QoL, the study group had significantly better scores than the control group (Table 3). At the end of the trial, 30 (66.7%) of the patients had at least a 50% improvement in their total PAC-QoL score (8 from the control group and 22 from the study group; *p* = 0.001) and 29 (64.4%) had reported ≥3 CBMpW. In the study group, 84% (*n* = 21) of the patients had CBMpW ≥ 3 compared to 40% (*n* = 8) in the control group (*p* = 0.003).

## 4. Discussion

Gut microbiota are affected by changes in the diet. Consuming more fiber in the diet results in higher quantities of *Provotella* spp. in the colon, whereas more protein and fat consumption cause *Bacteroides* spp. to reproduce, causing maladjustment of the gut microbiota, which leads to changes in nutrient absorption, immune response, and tolerance to symbiotic bacteria [21,22].

In a non-randomized controlled study evaluating features of fecal flora in FC patients, it was determined that *Bifidobacterium* and *Bacteroides* species were significantly low in stool samples of patients with FC [23]. The mean Bristol Stool Scores and CBMpW were significantly improved after a 2-week probiotic treatment. In another pivotal cross-sectional study conducted on children with constipation using 16S rRNA gene pyrosequencing, it was determined that *Prevotella* was abundant with several genera of *Firmicutes* in constipated patients compared to controls [24]. It was interpreted that the changes in the microbiome were due to a low-fiber diet, and bacterial fermentation end-products, such as increased butyrate production, might lead to constipation.

Increased fiber intake is a key principle in FC therapy. The physicochemical properties of fiber have a significant effect on the gut microbiota. The type of dietary fiber consumed affects the gut microbiota because not all types of bacteria have the capacity to produce the enzymes necessary for their digestion [25]. In the guidelines, soluble fibers are recommended for the treatment of constipation because there may be tolerance problems with insoluble fibers (e.g., fiber in wheat brans and whole grains) in some patients [26]. Insoluble fibers may lead to or increase abdominal pain, distention, and flatulence. Fruit fiber (e.g., prunes) or mixed soluble fibers are shown to be more effective in the short term than psyllium. Also, oligofructose-probiotic combinations are shown to have significant effects on chronic constipation [22]. In this study, patients on the study arm have achieved significant improvement in 6-week treatment with the personalized diet. Most of the patients on the customized diet were satisfied with the treatment approach, and both the number of CBMpW and the ratio of patients with more than 50% improvement in defecation frequency increased.

Considering the fact that nutrition alters the gut microbiota significantly, it is important to prepare a proper diet for patients with FC according to their needs. In our study, we have determined that personalized microbiome modulation by dietary intervention based on AI-assisted fecal microbiome profiling resulted in improvements in the symptoms of FC patients as well as their quality of life.

There are some limitations to the study. As a single-center pilot study, the results cannot be generalized to the whole patient population with FC. Also, there was no follow-up period after six weeks, so the waxing of symptoms, if any, has not been recorded. Lastly, due to financial limitations, microbiome tests have only been applied to study group patients instead of all the patients in the study.

In conclusion, customization of a diet based on individual microbiome tests provides better outcomes both clinically and socially in FC patients. Considering the significant social impact and healthcare costs related to FC, effective non-pharmacological therapies should be preferred for these patients. To our knowledge, this is the first study to utilize personalized dietary modulation intervention based on individual microbiome profiles of the FC patient population in Turkey and the literature.

## Figures and Tables

**Figure 1 jcm-11-06612-f001:**
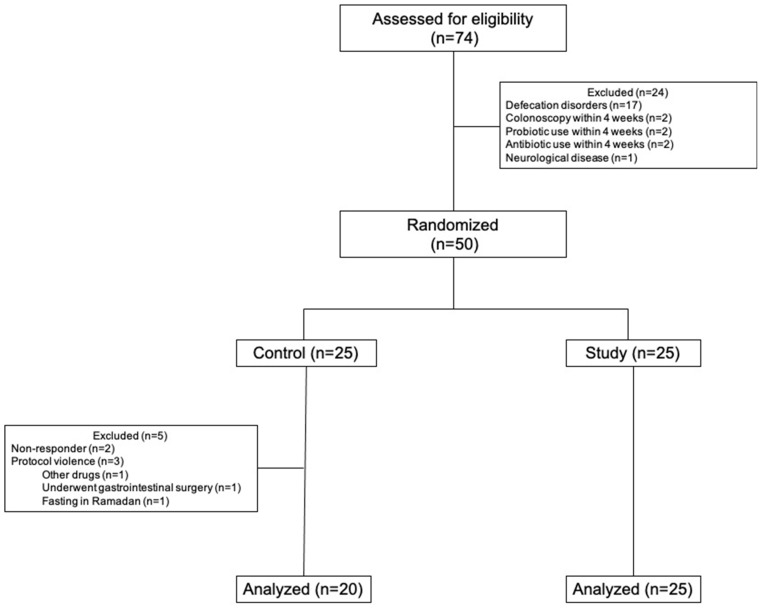
Flow diagram of the study.

**Table 1 jcm-11-06612-t001:** Demographic and clinical characteristics of the patients, baseline stool frequency, and quality of life scores.

Variables	Total (*n* = 45)	Control Group (*n* = 20)	Study Group (*n* = 25)	*p*
**Age** (years, mean ± SD)	31.5 ± 10.2	34.5 ± 11.4	29.1 ± 8.6	0.76 *
**Gender**				0.608 **
Male	5 (11.1)	2 (10)	3 (12)	
Female	40 (88.9)	18 (90)	22 (88)	
**BMI** (kg/m^2^ mean ± SD)	26.3 ± 5.1	26.1 ± 5	26.5 ± 5.3	0.786 *
**Constipation duration** (months, mean ± SD)	88.8 ± 66.9	91.9 ± 75.9	86.2 ± 60.2	0.778 *
**CBMpW** (*n*, mean ± SD)	1.9 ± 1.92	2.1 ± 2.2	1.7 ± 1.6	0.374 ***
**CBMpW** ≥ 3 (*n*, %)	6 (13.3)	4 (20)	2 (8)	0.383 *
**PAC-QoL subscales** (points, mean ± SD)				
Physical discomfort	10.33 ± 2.5	10 ± 2.4	10.5 ± 2.5	0.494 *
Psychosocial discomfort	17.33 ± 5.3	20.3 ± 4	15 ± 5.1	0.001 *
Worries and discomfort	30.8 ± 9.4	32.3 ± 5.8	29.6 ± 11.4	0.314 ***
Satisfaction	3.2 ± 2.1	3.3 ± 2.2	3.1 ± 2.1	0.736 *
**Total PAC-QoL** score (points, mean ± SD)	55.3 ± 14.6	59.3 ± 10.4	52.1 ± 16.9	0.101 *

SD: Standard deviation, BMI: Body mass index, CBMpW: Complete bowel movement per week, PAC-QoL: Patient Assessment Constipation–Quality of Life, *: Student’s *t* test, **: Pearson chi-square test, ***: Mann-Whitney-U test.

**Table 2 jcm-11-06612-t002:** Effect of treatments on stool frequency and quality of life at baseline and post-treatment.

	Control Group				Study Group			
	Baseline	After 10 Weeks	*T*	*p **	Baseline	After 10 Weeks	*T*	*p **
**CBMpW** (*n*, mean ± SD)	2.1 ± 2.2	2.8 ± 2	−3.462	**0.003**	1.7 ± 1.6	4.3 ± 1.8	−10.718	<0.001
**PAC-QoL** (points, mean ± SD)								
Physical discomfort	10.1 ± 2.4	9.8 ± 2.3	0.677	0.506	10.6 ± 2.5	5 ± 3.9	6.551	<0.001
Psychosocial discomfort	20.3 ± 4	19.4 ± 3.5	1.294	0.211	15 ± 5.1	6.5 ± 5.3	6.987	<0.001
Worries and discomfort	32.3 ± 5.9	29.8 ± 5.7	2.708	**0.014**	29.6 ± 11.5	15.2 ± 8.1	6.982	<0.001
Satisfaction	3.3 ± 2.2	3.9 ± 2.3	−1.332	0.199	3.1 ± 2.1	10.7 ± 3.5	−9.553	<0.001
**Total PAC-QoL score** (points, mean ± SD)	59.3 ± 10.4	55 ± 8.5	3.155	**0.005**	52.1 ± 16.9	15.9 ± 16	9.317	<0.001

CBMpW: Complete bowel movement per week, SD: Standard deviation, PAC-QoL: Patient Assessment Constipation Quality of Life, *: Paired samples *t*-test. Bold characters were used for statistically meaningful *p* values.

**Table 3 jcm-11-06612-t003:** Comparison between groups in terms of post-treatment stool frequency and quality of life measures.

Variables	Total (*n* = 45)	Control Group (*n* = 20)	Study Group (*n* = 25)	*p*
**CBMpW** (*n*, mean ± SD)	3.6 ± 2	2.8 ± 2	4.3 ± 1.8	0.013 *
**PAC-QoL** (points, mean ± SD)				
Physical discomfort	7.1 ± 4.1	9.8 ± 2.3	5 ± 3.9	<0.001 *
Psychosocial discomfort	12.2 ± 7.9	19.3 ± 3.5	6.5 ± 5.4	<0.001 **
Worries and discomfort	21.7 ± 10.2	29.9 ± 5.6	15.2 ± 8.1	<0.001 *
Satisfaction	7.7 ± 4.5	3.9 ± 2.3	10.7 ± 3.5	<0.001 **
**Total PAC-QoL score** (points, mean ± SD)	33.3 ± 23.6	55.1 ± 8.5	15.9 ± 16	<0.001 **
**50% improvement in total score** (*n*, %)	30 (66.7)	8 (40)	22 (88)	0.001 ***
**CBMpW ≥ 3** (*n*, %)	29 (64.4)	8 (40)	21 (84)	0.003 ***

CBMpW: Complete bowel movement per week, SD: Standard deviation, PAC-QoL: Patient Assessment Constipation–Quality of Life. *: Student’s *t* test, **: Mann-Whitney-U test, ***: Pearson chi-square test.

## Data Availability

The data presented in this study are available on request from the corresponding author. The data are not publicly available due to privacy reasons.

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
