# Peer review of "Efficacy of AI-Assisted Personalized Microbiome Modulation by Diet in Functional Constipation: A Randomized Controlled Trial"

_jcm, 2022, doi:10.3390/jcm11226612_

Round 1

Reviewer 1 Report

The manuscript is interesting, well-written, comprising all necessary sections, and addresses a frequently encountered pathology in daily practice - functional constipation. However, there are some issues that I recommend be solved by the authors.

1. In section 2, subsection 2.1 Study design you should mention the period and center where the study was conducted. In addition, a flowchart regarding the study design would add value to the manuscript.

2. It seems unclear what kind of fiber/food regarding AI-based nutrition was recommended. It would be helpful to detail this subject, mainly focusing on microbioma.

3. Did you study the microbioma of both groups? a comparative analysis would be interesting.

4. The diagnosis of functional constipation requires extensive investigations for the differential diagnosis. How did you exclude other pathologies that cause constipation? When was the colonoscopy performed at the latest?

Author Response

  1. In section 2, subsection 2.1 Study design you should mention the period and center where the study was conducted. In addition, a flowchart regarding the study design would add value to the manuscript.

Answer: The name of the institution was added to related section. The flow chart was added to results section. 

  1. It seems unclear what kind of fiber/food regarding AI-based nutrition was recommended. It would be helpful to detail this subject, mainly focusing on microbioma.

Answer: A relevant section is added to the sub-section and a reference to a previous publication is added.

  1. Did you study the microbioma of both groups? a comparative analysis would be interesting.

Answer: Unfortunately we did not study microbiome for both groups due to cost issues. We mentioned this limitaion in discussion.

  1. 4. The diagnosis of functional constipation requires extensive investigations for the differential diagnosis. How did you exclude other pathologies that cause constipation? When was the colonoscopy performed at the latest?

Answer: All these details can be found in "Materials and Methods" section (line 93-105). We performed colonic transit time and MR dephecography for all eligible patients and excluded other causes of constipation. Patients with a colonoscopy within last 4 weeks were excluded however, colonoscpoy within last 5 years was an inclusion criterion.

Reviewer 2 Report

Dear authors, its quite an interesting topic and valuable findings. 

please see some comments for the improvement of manuscript 

Introduction: is well written 

Methods: Methodology is also well described, however, did the authors took permission to use ROME diagnostic tool?

Sample size: Which tool/software was used to calculate sample size? please cite the reference. 

Results: Table 1 and 3, please change the heading n(%) to parameters or some other suitable wording. as n(%) or Mean SD is already stated with your variables. 

all the best for your paper. 

Author Response

Methods: Methodology is also well described, however, did the authors took permission to use ROME diagnostic tool?

Answer: The necessary license has been obtained from the ROME Foundation.

Sample size: Which tool/software was used to calculate sample size? please cite the reference. 

Answer: A web-based Sample Size Calculator by Raosoft, Inc. was used. It's added in the references.

Results: Table 1 and 3, please change the heading n(%) to parameters or some other suitable wording. as n(%) or Mean SD is already stated with your variables. 

Answer: n (%) was replaced with "variables".

Round 2

Reviewer 1 Report

The authors have revised the manuscript accordingly.